# A New Gnetalean Macrofossil from the Mid-Jurassic Daohugou Formation

**DOI:** 10.3390/plants12091749

**Published:** 2023-04-24

**Authors:** Yong Yang, Zhi Yang, Longbiao Lin, Yingwei Wang, David Kay Ferguson

**Affiliations:** 1Co-Innovation Center for Sustainable Forestry in Southern China, College of Biology and the Environment, Nanjing Forestry University, 159 Longpan Road, Nanjing 210037, China; 2Independent Researcher, 69 Fuxing Road, Beijing 100039, China; 3National Botanical Garden, Institute of Botany, Chinese Academy of Sciences, 20 Nanxincun, Xiangshan, Beijing 100093, China; 4Department of Paleontology, University of Vienna, 1090 Vienna, Austria

**Keywords:** China, Daohugou Formation, *Daohugoucladus*, evolution, gnetophytes, Inner Mongolia

## Abstract

Macrofossil evidence has demonstrated a first radiation of gnetophytes in the Early Cretaceous. However, the origin of the diversity of gnetophytes remains ambiguous because gnetalean macrofossils have rarely been reported from pre-Cretaceous strata. Here, we report a new putative gnetalean macrofossil reproductive shoot which possesses opposite phyllotaxy, long linear leaves more or less decurrent and having a prominent midvein and pedicled ovoid-ellipsoid and longitudinally striated chlamydosperms. Our new fossil is different from other known gnetalean macrofossils in the linear-lanceolate leaves with a midvein and pedicled chlamydosperms. As a result, we describe this new macrofossil reproductive shoot as new to science, i.e., *Daohugoucladus sinensis* gen. et sp. nov. Our new macrofossil displays additional morphological characters distinct from other known Mesozoic and modern gnetalean species and provides additional evidence of the origin and early evolution of female reproductive organs of gnetophytes.

## 1. Introduction

Gnetophytes contain three extant monotypic families (Ephedraceae, Gnetaceae and Welwitschiaceae), which possess unusual morphological characters for seed plants, e.g., bisexual cones, flower-like reproductive organs, style-like micropylar tube, unique chlamydosperms with additional envelope(s) partially enclosing the inner ovule and exposing an apical micropylar tube, vessels in the wood anatomy, double fertilization and archegonia lacking in Gnetaceae and Welwitschiaceae [1,2,3,4,5,6,7,8,9]. Since these transitional characters bridge the gap between other gymnosperm groups and angiosperms, emphasis has been placed on this group to seek clues to the origin of angiosperms, and a number of competing hypotheses have been proposed, e.g., euanthial hypothesis, pseudanthial hypothesis, anthophyte hypothesis, neo-pseudanthial hypothesis [10]. Recent phylogenomic and phylotranscriptomic studies have suggested that gnetophytes constitute a monophyletic group which is sister to Pinaceae [11,12]. Despite these advances, the evolutionary history of the unusual morphological characters of gnetophytes has not been critically evaluated.

Paleobotanical studies can provide convincing evidence of the evolutionary history of organisms. Numerous gnetalean macro- and mesofossils have been reported from the Mid-Jurassic to Early Cretaceous, i.e., Asia [13,14,15,16,17,18,19,20,21,22,23,24,25,26,27,28,29,30,31], Australia [32], Europe [33,34,35,36,37], North America [33,34,35,36,38,39,40] and South America [41,42,43,44,45,46,47,48,49,50]. These fossils show a wide distribution range (Australia, NE China, Mongolia, United States of America, Brazil and Portugal) and huge morphological diversity [22,23,25,27,33,35,42,50]. In the Early Cretaceous, Ephedraceae occurred in Oceania (Australia), Asia (Northeast China and Mongolia), southern Europe (Portugal) and South America (Brazil) [22,23,24,28,29,32,33]; Welwitschiaceae lived in southern Europe, North America and South America [35,41,42]; and Gnetaceae was apparently restricted to Asia (Northeast China) [27]. Thus far, more than 54 species have been recorded from the Mesozoic. Most of these macrofossils were reported after 1996 and laid the foundation for our understanding of the diversity and evolution of early gnetophytes.

The enormous diversity is of great significance in unveiling the evolutionary process of female reproductive organs in ephedroid plants. Chlamydosperms are characteristic of gnetophytes, possessing one or two layers of cupules enclosing an inner ovule with a style-like micropylar tube exposed, while female cones of gnetophytes consist of a number of bracts and their axillary chlamydosperms [1,4]. The chlamydosperm of gnetophytes was hypothesized to have originated from a reduced secondary shoot with a basal pair/whorl of foliar organs fused into a cupule enclosing the inner ovule, according to morphology, anatomy, ontogeny and paleobotany [7,8,39,51]. Female cones of gnetophytes have been thought to be compound and homologous to those of conifers [2,51]. Indeed, the female cone of Protoephedraceae is compound and the secondary fertile shoots possess sporophylls distally and bracteoles proximally; Protoephedraceae possessed exposed ovules but no chlamydosperms and were thought to represent the ancestral lineage of eugnetophytes [39]. The origin and evolution of the compound female cone of eugnetophytes were not elucidated until those macrofossils from the Early Cretaceous were discovered recently. A reduction–sterilization evolutionary model has been proposed to explain the origin and evolution of ephedroid female cones: the chlamydosperm is pedicled after its origin and subsequent changes during the evolutionary process include the chlamydosperm losing its pedicel and becoming sessile in the axils of leaves, internodes of the female spike increasingly shortened, leaves modified into bracts and reduction of the lower pairs of bracts [19,31].

However, how this morphological diversity in the Early Cretaceous originated remains arguable due to the lack of macrofossils from earlier geological strata (e.g., the Jurassic). Macrofossil taxa of gnetophytes were rarely found from the Jurassic [40,52]. Here we report a new macrofossil taxon from the Mid-Jurassic Daohugou Formation which provides additional morphological evidence for a better understanding of the evolutionary history of gnetalean plants.

## 2. Results

Gnetidae

*Daohugoucladus sinensis*, gen. et sp. nov., Figure 1, Figure 2, Figure 3, Figure 4 and Figure 5.

Type: NF2020123101 (holotype: NF).

Diagnosis: A reproductive shoot having nodes and internodes; leaves opposite, linear to linear-lanceolate, subsessile and possessing a prominent midvein on both sides, decurrent; chlamydosperms terminal on a long and furrowed pedicel, ovoid to elliptic-ovoid, longitudinally striated, truncate or nearly so at the apex with a central projection.

Description: A fossil specimen containing compression of a reproductive shoot, ca. 7.1 cm long (Figure 1 and Figure 5). The unbranched shoot articulate, possessing nodes and internodes (Figure 2a,b). The nodes swollen. The internodes ca. 5 mm long or slightly shorter. Leaves opposite at nodes, linear to linear-lanceolate, 1–2 mm broad, 1.5–4.1 cm long, shorter proximally, longer distally; apex attenuate to lanceolate, base acute and decurrent, subsessile; midvein prominent, impressed on both surfaces (Figure 3a,b); fine longitudinal striations discernible at both sides of the midvein on the abaxial surface (Figure 3). Chlamydosperms pedicled; pedicel long and axillary to leaves, longitudinally furrowed, straight or slightly curved, ca. 2.2 cm long (Figure 4a). Chlamydosperms ovoid to elliptic-ovoid, 4–6 × 3–4 mm, having longitudinal ridges and furrows, base acute, apex nearly truncate, the apical portion elevated in the middle forming a micropylar region (Figure 1b and Figure 4b,c).

**Figure 1 plants-12-01749-f001:**
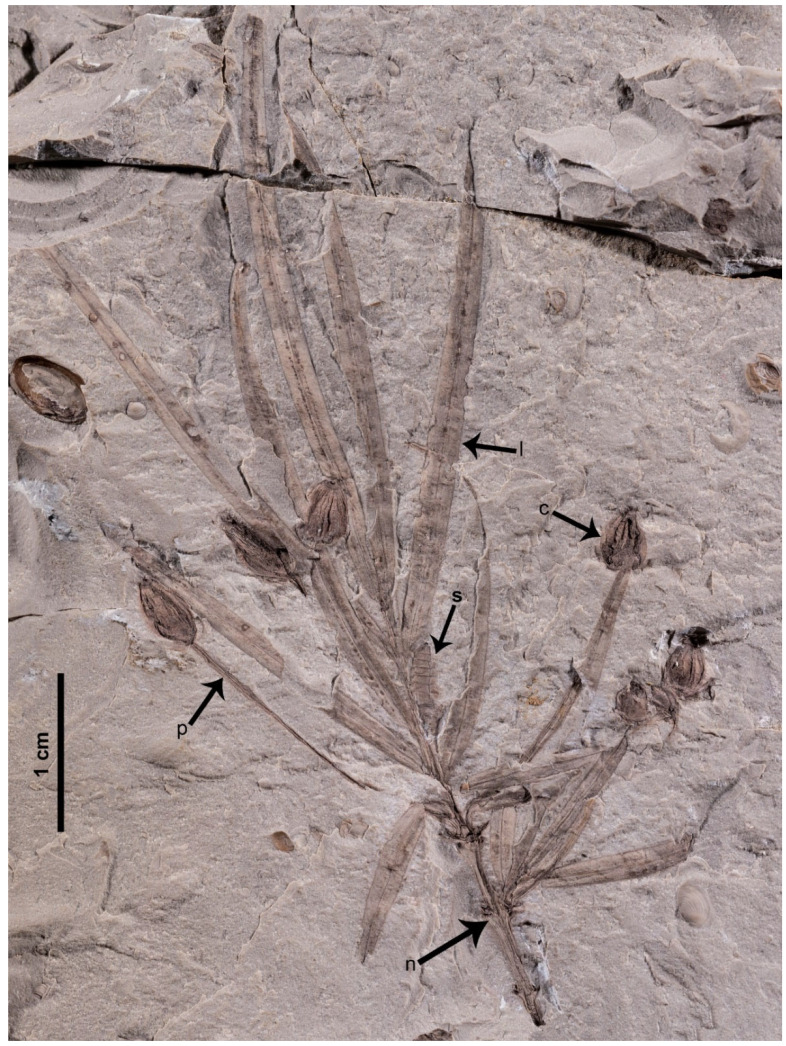
Holotype of *Daohugoucladus sinensis* gen. et sp. nov. displaying overall morphology including the opposite phyllotaxy, the lengthy linear leaves with prominent midvein and pedicled chlamydosperms axillary to leaves. Abbreviations: c, chlamydosperms; l, leaf; n, node; p, pedicel; s, insect.

Etymology: The generic name *Daohugoucladus* is derived from the geological stratum of the fossil (the Daohugou Formation) and the preservation of the reproductive shoot; the specific epithet *sinensis* means that the fossil is from China.

Type locality: Daohugou Village, Ningcheng County, Inner Mongolia, China.

Stratigraphy: The Daohugou Formation, Mid-Jurassic (~165 myr).

Remarks: The general morphology of both vegetative and reproductive organs excludes the affinity of our new fossil to ferns and fern allies, cycads, *Ginkgo* L. and angiosperms, while showing similarity to conifers and gnetophytes. The opposite phyllotaxis is common in both extant and extinct gnetalean plants, while the long linear leaves are decurrent and possess a prominent midvein, as in conifers. Our new fossil shows a certain similarity to modern *Podocarpus* L’Hér. ex Pers., e.g., linear and midveined, but the reproductive morphology does not support such a relationship, e.g., the pedicled and orthotropous chlamydosperm lacking any subtending bracts (vs. reduced female cones usually with anatropous ovules and more or less fleshy receptacle and epimatium in *Podocarpus*). Our new fossil is markedly different from Jurassic or earlier conifers in having a single ovulate organ terminal to pedicels (vs. assembled into compound cones or shoots in earlier conifers) [53].

**Figure 2 plants-12-01749-f002:**
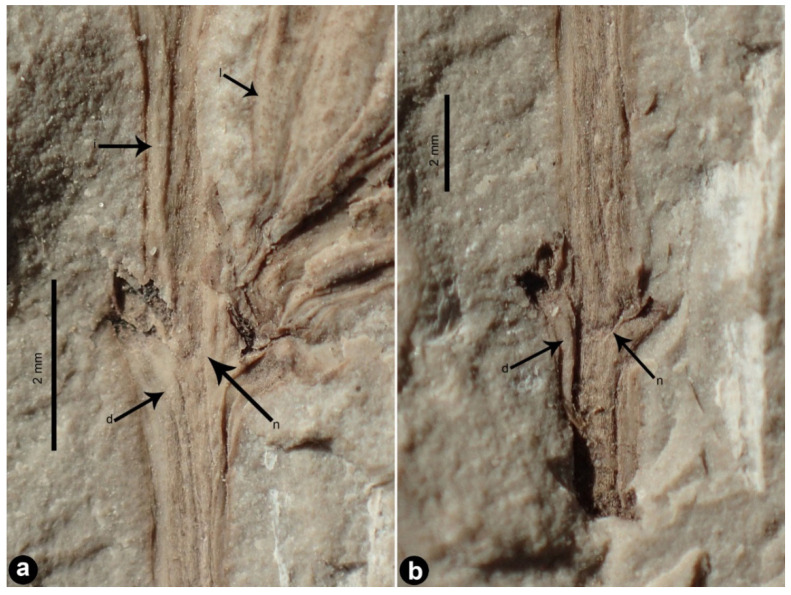
Shoot portion displaying nodal morphology of *Daohugoucladus sinensis* gen. et sp. nov. (**a**) middle node displaying opposite leaf position and decurrent leaf bases; (**b**) proximal node displaying decurrent leaf base. Abbreviations: d, decurrent leaf base; i, internode; l, leaf; n, node.

*Daohugoucladus* is similar to *Juraherba* Han et Wang in its leaf morphology (linear and midveined). *Juraherba* was described as an herbaceous angiosperm from the same stratum [54]. However, the two species are different from one another, e.g., leaves are spirally arranged in *Juraherba* but opposite in *Daohugoucladus*, fructifications/chlamydosperms are shorter in *Juraherba* (2.2–4.1 mm long) than in *Daohugoucladus* (4–6 mm long). Moreover, the pedicels of the fructifications/chlamydosperms are shorter in *Juraherba* (14–15.5 mm long) than in *Daohugoucladus* (ca. 22 mm long) and possess scaly leaves in *Juraherba* which are lacking in *Daohugoucladus*. In addition, *Juraherba* possesses bract-like structures at the base of the fructifications, while *Daohugoucladus* has no bract-like structures at the base of the chlamydosperms. These differences can clearly distinguish *Daohugoucladus* from *Juraherba*.

**Figure 3 plants-12-01749-f003:**
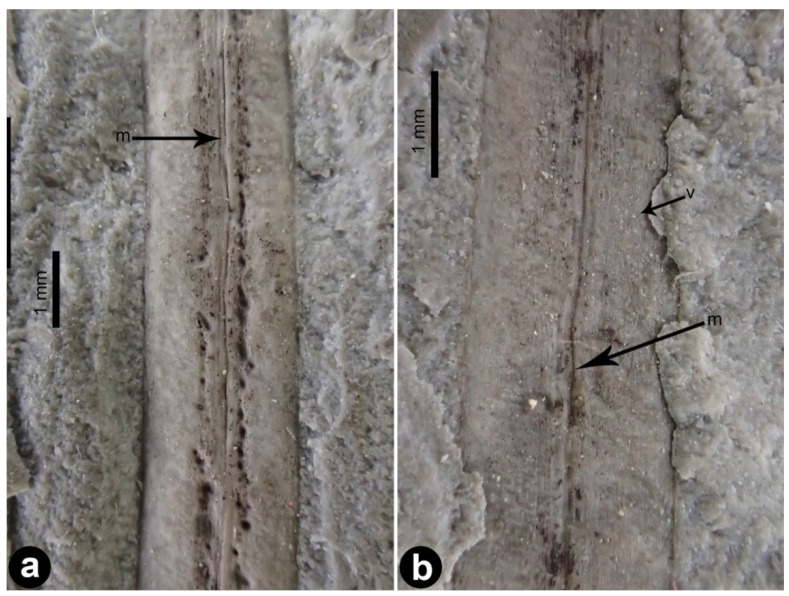
Leaf morphology of *Daohugoucladus sinensis* gen. et sp. nov. (**a**,**b**) leaf portion displaying the midvein and the longitudinal striations. Abbreviations: m, midvein; v, longitudinal striations.

*Daohugoucladus* resembles gnetalean plants such as *Siphonospermum* in the opposite phyllotaxy and the chlamydosperms terminal to pedicles but differs from gnetophytes in the leaf morphology. However, leaves of fossil gnetophytes are highly variable, e.g., linear in many ephedroid plants, broad with pinnate venation in *Gnetum* L. and *Constrobilus* H.M. Liu et al. [26], divided in *Latibractea divisa* H.M. Liu et al. [26], strap-shaped in *Ephedra multinervia* Y. Yang et L.B. Lin [20] and *Welwitschia mirabilis* Hook. f. [1]. As a result, we think that the midveined leaf shape alone cannot exclude its affinity to gnetophytes.

The chlamydosperms of *Daohugoucladus* are pedicled and not assembled into a compound female cone, which reminds us of the female organ of *Siphonospermum* Rydin et Friis, an ephedroid macrofossil from the Early Cretaceous of the Yixian Formation [28]. *Daohugoucladus* differs from *Siphonospermum* in the lengthy linear leaves with a prominent midvein and minor longitudinal striations, and the chlamydosperms having a shorter micropylar tube as a central projection (vs. leaves with parallel veins lacking a midvein, and the chlamydosperm having a longer micropylar tube with an exposed portion 2.5–3.5 mm long in *Siphonospermum*). The overall shape of the chlamydosperms in our new fossil also resembles gnetalean chlamydosperms from the Early Cretaceous [34,36], but our fossil differs from those Early Cretaceous mesofossil seeds in the seed envelope being undivided and having longitudinal striations (vs. usually three- or four-parted and seed envelope having transverse ridges) [34,36].

The chlamydosperms of *Daohugoucladus* are ovoid to elliptic-ovoid, which is similar to that of modern *Gnetum*, but markedly different from those bilateral chlamydosperms in modern *Ephedra* L. and *Welwitschia* Hook. f. However, the chlamydosperms of *Daohugoucladus* are solely positioned at the top of a long pedicel, and not verticillately arranged at nodes and assembled into loose spikes (vs. sessile, verticillately arranged at nodes and assembled into loose spikes in modern *Gnetum*; refer to Pearson [1] and Maheshwari and Vasil [55]. Our new fossil does not show close similarity to modern gnetophytes except for the opposite phyllotaxy and the presence of the unusual chlamydosperms.

**Figure 4 plants-12-01749-f004:**
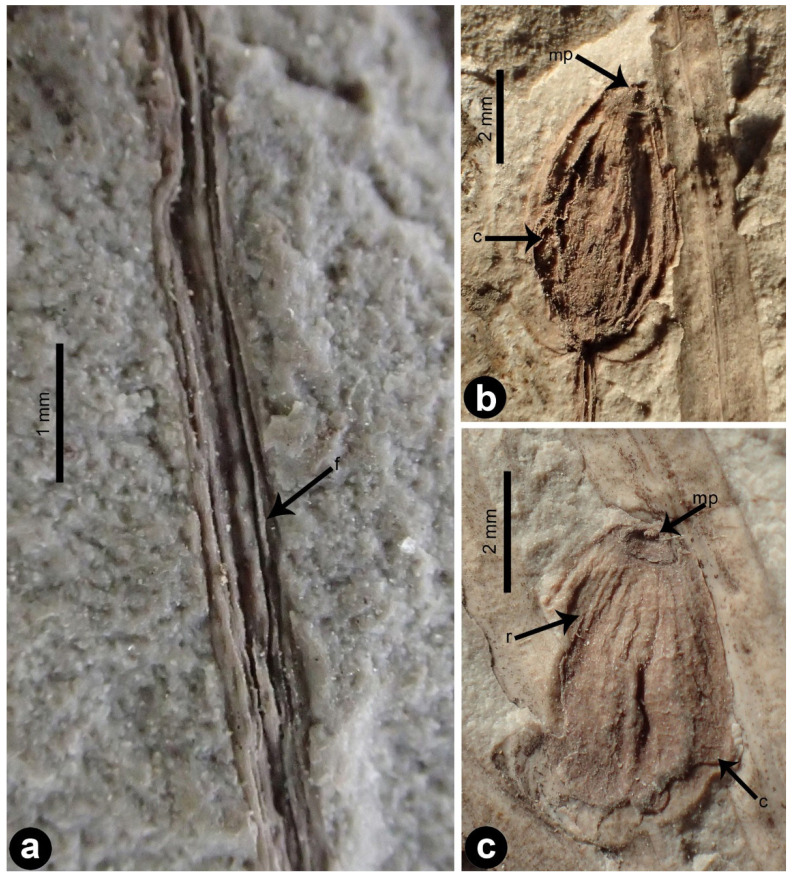
Reproductive morphology of *Daohugoucladus sinensis* gen. et sp. nov. (**a**) pedicle portion displaying the longitudinal furrows and ridges; (**b**,**c**) chlamydosperms displaying the ovoid to ellipsoid shape and the surface longitudinal ridges, the apical portion with a central projection. Abbreviations: c, chlamydosperm; f, furrow; mp, micropylar tube projection; r, ridge.

Gnetophytes have chlamydosperms with an exposed micropylar tube [4,8]. Our new macrofossil is remarkable in that the chlamydosperm has an apical micropylar region and an extremely short micropylar projection, as also found in many gnetalean chlamydosperms from the Early Cretaceous [36], but unknown in other gymnosperm groups [56]. In addition, the micropylar projection of our new fossil is extremely short and appears truncate. In modern gnetophytes, the opening of the micropylar tube is usually not truncate but oblique [57]. The micropylar tube of modern gnetophytes is fragile and easily broken (pers. observ.) and broken micropylar tubes were frequently found in the Early Cretaceous mesofossil *Rothwellia foveata* Friis et al. [36]. As a result, it is reasonable to infer that the lack of a prominent micropylar tube is the result of breakage.

In the middle portion of the specimen, there is an articulate cylindrical structure adnate to the main shoot and seemingly axillary to the right leaf (Figure 1: s). We believe this structure is the remains of some Arthropoda species, and not organically connected with our new macrofossil plant. We came to this conclusion for a number of reasons: (1) The organic connection between this cylindrical structure and the fossil plant is ambiguous. It is highly probable that they were just preserved together. (2) There is only one such cylindrical structure preserved together with the fossil plant. (3) It is clear that the fossil plant possesses female reproductive organs and the cylindrical structure would have to be a male reproductive organ if it belongs to the fossil plant. While some may argue that it is similar to the verticillately arranged male cone of *Gnetum*, it is implausible to have such a strange fossil plant possessing very primitive female reproductive organs like that of *Siphonospermum*, but with highly specialized male cones similar to modern *Gnetum* L. on the other hand.

**Figure 5 plants-12-01749-f005:**
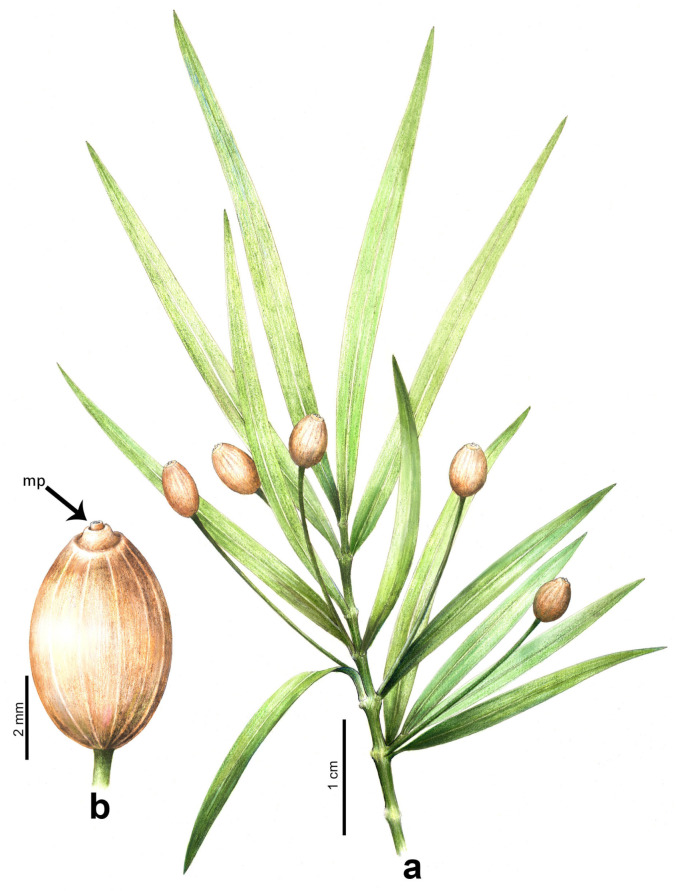
Line drawing of *Daohugoucladus sinensis* gen. et sp. nov. (**a**) reproductive shoot displaying the articulate shoot having swollen nodes, opposite linear leaves, with pedicled chlamydosperms axillary to leaves; (**b**) ellipsoidal chlamydosperm with apical micropylar tube projection. Abbreviations: mp, micropylar tube projection. Line-drawing: Aili Li.

The unusual combination of vegetative and reproductive morphology suggests that this macrofossil belongs to a gnetalean genus that has not yet been described. As a result, we classified the new gnetalean plant as a Gnetidae according to Christenhusz [58] and describe it here as new to science.

## 3. Materials and Methods

One fossil specimen was collected from the fossiliferous bed in Daohugou Village, Shantou Township, Ningcheng County, Inner Mongolia Autonomous Region of China, which is near the border of Liaoning Province and Inner Mongolia (Figure 6). The fossil specimen was a compression with coalified residues embedded in tuffaceous siltstone. The Daohugou Formation contained a Mid-Jurassic flora [54]. Many fossils of plants and animals were found in this fossil bed [52,54,59]. The absolute age of this fossiliferous bed was dated back to the Mid-Jurassic (~164 myr, Callovian) based on Ar40/Ar39 and SHRIMP U-Pb dating methods [52,54,60,61]. This new fossil was studied using a light microscope and digital cameras (Nikon D850, D7100, Olympus TG4 and Sony A7M3). The fossil specimen is deposited in the Herbarium (NF), Nanjing Forestry University, Nanjing, Jiangsu Province of China.

## 4. Discussion

Phyllotaxy is conservative within the gnetophytes. The three living families of gnetophytes usually possess decussate phyllotaxy [1,4], but sometimes ternately whorled phyllotaxy does occur in *Ephedra* [8,62,63]. Most gnetalean macrofossils do have decussate phyllotaxy, i.e., opposite leaves or branching pattern (Table 1) [18,19,20,21,22,23,24,28,29,38,50]; *Alloephedra xingxuei* J.R. Tao et Y. Yang is considered as having alternate leaves, which is arguable due to the poor preservation of vegetative characters [17,25]. Our new macrofossil reported here shows that the gnetophytes possess decussate phyllotaxy, so it is reasonable to infer that the decussate or ternately whorled phyllotaxy is conservative and a synapomorphic character of the gnetophytes.

Mesozoic macrofossils have shown additional diversity of leaf morphology in the gnetophytes. The three extant families of gnetophytes displays rather stable leaf morphology. Species of Welwitschiacae possess giant strap-shaped leaves with no midvein but multiple parallel veins (Table 1) [1,4]. This leaf pattern was also found in the macrofossils having a Welwitschiaceae affinity from the Early Cretaceous [38,41,42]. Species of Gnetaceae have elliptic to ovoid broad leaves that are pinnately veined, i.e., there is a midvein and a few pairs of lateral veins [1,4,55]. Macrofossils with a Gnetaceae affinity are rare. *Khitania* S.X. Guo et al. is a male spike from the Yixian Formation, and it preserved no vegetative characters [27]. *Protognetum* Y. Yang et al. from the Mid-Jurassic Daohugou Formation is a reproductive shoot and possesses ephedroid vegetative characters (e.g., opposite phyllotaxy and linear and parallel-veined leaves) and verticillate sessile chlamydosperms [52]. In Ephedraceae, taxa have opposite or ternately whorled, sessile, linear leaves that are free or fused at the base, forming a sheath with two or three free apical parts; the length of the sheath and the apical free portion is variable among species and normally each leaf has two parallel veins and there is no midvein [1,4]. Early Cretaceous ephedroid macrofossils have shown much higher diversity of leaf morphology than the modern representatives [20,24,25,26,28,31]. Some of the Cretaceous ephedroid macrofossils possess no leaves, e.g., *Ephedra hongtaoi* Wang et Zheng [29] and *Chengia laxispicata* Y. Yang et al. [19]; some have linear, free and parallel-veined leaves, e.g., *Liaxia* Cao et Wu [24] and *Prognetella* Krassilov [31]; some others, however, are provided with petiolate broad leaves with pinnate veins (*Constrobilus ovata* H.M. Liu et al.) [26], or petiolate divided leaves with parallel veins (*Latibractea divisa* H.M. Liu et al.) [26], or petiolate ovate leaves with forked venation (*Spinobractea lanceoleta* H.M. Liu et al.) [26], or even strap-shaped leaves with parallel veins similar to those of Welwitschiaceae (*Ephedra multinervia* Y. Yang et L.B. Lin) [20]. Our new macrofossil in this study has decurrent, long, linear leaves with a prominent midvein and multiple longitudinal striations. The midveined linear leaves of our new fossil show a certain resemblance to that of modern *Podocarpus*. However, the female reproductive organ of our new fossil is quite different from that of *Podocarpus* in the pedicled orthotropous ovulate organs (vs. reduced female cones with a basal enlarged receptacle and terminal anatropous ovulate organs) [66]. The leaf pattern of our new fossil is new to gnetophytes and has not been reported in gnetalean fossils. This Mid-Jurassic macrofossil unveils additional leaf diversity of ephedroid plants and furthers our understanding of the early diversification of gnetophytes.

Modern families of gnetophytes have diversified female cones or spikes: Welwitschiaceae possess typical female cones with sessile winged chlamydosperms axillary to bracts [1,4]; Gnetaceae have female spikes with a number of annular structures of modified bracts subtending axillary verticillate chlamydosperms [55]; and Ephedraceae are provided with extremely reduced female cones with only the uppermost pair/whorl of bracts subtending 1–3 sessile chlamydosperms [1,4,5,6,19]. The female cones/spikes of all three modern families of gnetophytes are thought to be compound and consist of a number of modified bracts subtending axillary chlamydosperms [1,4,5]; the chlamydosperms are specialized reproductive shoots with a basal pair of foliar bracteoles fused into the cupule-like envelope enclosing the inner ovule according to morphological, tetratological, anatomical and ontogenetic evidence [7,8,51,67].

Paleobotanical evidence is pivotal for clarifying the early evolution of female reproductive organs in gnetophytes. *Protoephedrites* Rothwell et Stockey is a well-preserved macrofossil from the Early Cretaceous and has a compound female reproductive shoot with decussate bracts subtending axillary fertile shoots; the secondary fertile shoots possess one or two pairs of bracteoles and a pair of orthotropous ovules with a short micropylar tube [39]. This fossil displays the hypothesized characters of ancestral gnetophytes and has no typical chlamydosperms but a secondary fertile shoot (Figure 7a) [7,8,19,51], providing solid evidence for the shoot origin hypothesis of chlamydosperms of eugnetophytes. *Protoephedrites* represents a continuation of stem gnetophytes in the Early Cretaceous. Furthermore, many other Early Cretaceous fossils have demonstrated the first radiation of gnetophytes on earth [68,69], and revealed huge diversity of female cones showing how the chlamydosperms of ephedroid plants were assembled into female spikes/cones and then how these female spikes/cones finally gave rise to the reduced female cones of *Ephedra* through reduction and modification [19,31,70]. The pedicled chlamydosperm of *Siphonospermum* represents the primitive form of eugnetophytes (Figure 7b). Later the chlamydosperms became sessile and the internode of the reproductive shoot became progressively shorter, forming loosely arranged female spikes as in *Protognetella*, in which the bracts were still leaf-like (Figure 7c). Subsequently, leaf-like bracts were modified into broad and short bracts tightly subtending the axillary chlamydosperms, the internodes of those loose female spikes were further shortened, forming increasingly compact female cones with multiple pairs of fertile bracts as in *Chengia*, *Liaoxia* and *Ephedra cantata* Puebla et al. (Figure 7d). Finally, the lower pairs of bracts of these compact female cones became abortive and only the uppermost pair of bracts remained fertile, forming reduced female cones as in *Ephedra* (Figure 7e) [19,31,70]. The lax female spike of Gnetaceae (Figure 7f) and the compact female cone of Welwitschiaceae (Figure 7g) may have been modified from the female shoot of *Prognetella*. These basic patterns of fossil ephedroid plants support the reduction and abortion evolutionary hypothesis [19].

## 5. Conclusions

A new gnetalean macrofossil, *Daohugoucladus sinensis* Y. Yang et al., is described from the Mid-Jurassic Daohugou Formation. The new fossil includes a reproductive shoot with long and decussate leaves and pedicled chlamydosperms. *Daohugoucladus sinensis* is similar to *Siphonospermum simplex* Rydin et Friis in the pedicled chlamydosperms, but differs from the latter in the long and mid-veined leaves. Both *Daohugoucladus sinensis* and *Siphonospermum simplex* display the primitive reproductive morphology of gnetophytes in the form of pedicled chlamydosperms. The finding of *Daohugoucladus sinensis* is important for a better understanding of the early evolution of gnetophytes because it extends the age of this primitive morphology back to the Mid-Jurassic.

## Figures and Tables

**Figure 6 plants-12-01749-f006:**
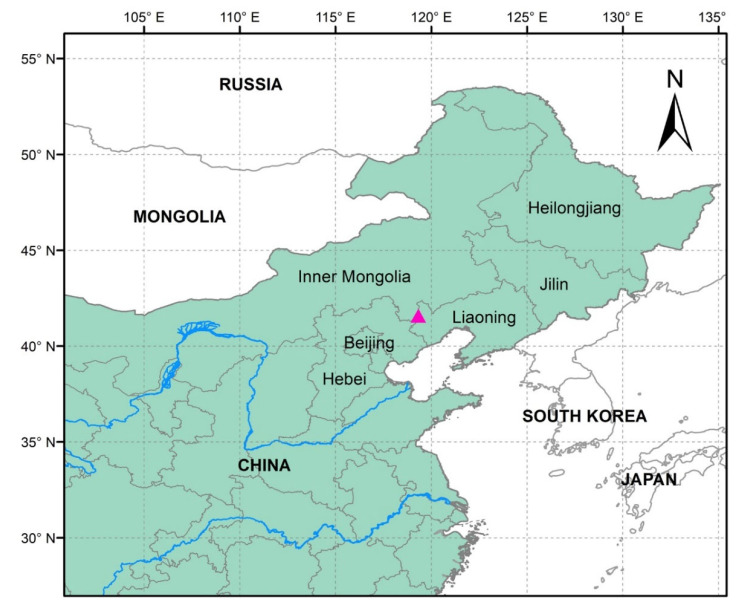
Map displaying the type locality of *Daohugoucladus sinensis* gen. et sp. nov. (pink solid triangle).

**Figure 7 plants-12-01749-f007:**
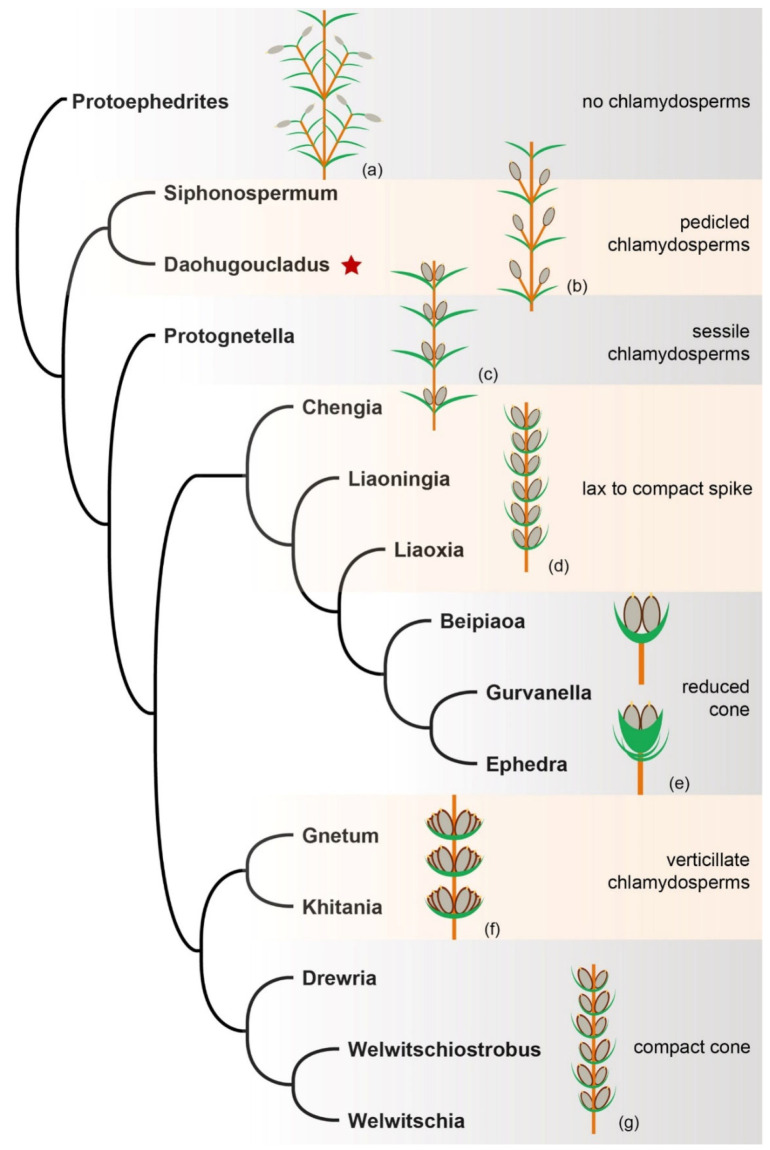
Supposed relationships among fossil gnetophytes with illustrations showing the reduction–sterilization evolutionary model of female cones of ephedroid plants. (**a**) ancestral female reproductive shoot displaying megasporophylls with apical naked othotropous ovule (e.g., *Protoephedrites* [39]); (**b**) transitional stage displaying the reproductive shoot with pedicled chlamydosperms (e.g., *Siphonospermum* [28] and *Daohugoucladus* gen. nov.); (**c**) transitional stage displaying the female shoot with sessile chlamydosperms axillary to leaves (e.g., *Prognetella* [31]); (**d**) transitional stage displaying the spike becoming compact with shortened internodes and modified bracts (e.g., *Chengia* [19], *Liaoningia* [21] and *Liaoxia* [24]); (**e**) reduced female cone with only the uppermost pair of bracts fertile (e.g., *Ephedra* [1] and *Gurvanella* [16]); (**f**) lax spike of Gnetaceae with verticillately arranged chlamydosperms [1]; (**g**) compact female cones of Welwitschiaceae with multiple whorls of chlamydosperms [1]. Red star indicates the new fossil plant described in this paper.

**Table 1 plants-12-01749-t001:** A morphological comparison between *Daohugoucladus* gen. et sp. nov. and a few extant and extinct gnetalean fossils.

Group	Taxon	Age	Leaf Position	Leaf Shape	Leaf Midvein Present	Leaves Decurrent	Leaf Petiole	Chlamydosperm Pedicled	Female Cone Organization	Micropylar Tube Prominent
Group I: pedicled chlamydosperms	*Daohugoucladus* gen. nov.	Mid-Jurassic	opposite	lengthy, linear	yes	yes	sessile	yes	chlamydosperms not assembled into cones	not prominent, but having central projection
	*Siphonospermum* [28]	Early Cretaceous	opposite	linear	no	no	sessile	yes	chlamydosperms pedicled, not assembled into spikes or cones	yes, 2.5–3.5 mm
Group II: reproductive shoots; sessile chlamydosperms; bracts leaf-like	*Prognetella* [20]	Early Cretaceous	opposite	linear	no	no	sessile	no	reproductive shoots with chlamydosperms axillary to leaf-like bracts	yes
Group II: reproductive shoots to female spikes or compact cones; internode shortened; sessile chlamydosperms; bracts modified	*Chengia* [19]	Early Cretaceous	opposite	linear	no	no	sessile	no	loose spikes with multiple pairs of modified fertile bracts	yes, 0.4–0.7 mm
	*Liaoningia* [21]	Early Cretaceous	opposite	strap-shaped	no	no	sessile	no	spike-like with multiple pairs of modified bracts each subtending an axillary sessile chlamydosperms	yes, 0.4 mm
	*Liaoxia* [24]	Early Cretaceous	opposite	linear	no	no	sessile	no	typical compact cones with multiple pairs/whorls of modified fertile bracts	yes
	*Dayvaultia* [40]	Late Jurassic	-	-	-	-	-	no	compact cones with fused bracts forming cupules and each bract subtending paired chlamydosperms	minute apical projection
	*Quadrispermum* (34]	Early Cretaceous	-	-	-	-	-	-	probably compact cones with sessile chlamydosperms axillary to modified bracts; ellipsoidal to ovoid chlamydosperms four-angled and having transverse ridges	micropylar tube present but probably broken
Group III: compact cones; sessile chlamydosperms; bracts modified, the lower pairs/whorls sterile, only the uppermost pair/whorl enclosing 1-3 chlamydosperms	*Spinobractea* [26]	Early Cretaceous	opposite	broadly lanceolate	no, but forked venation	no	petiolate	no	compact female cones with lengthy bracts enclosing 1–2 seeds	-
	*Constrobilus* [26]	Early Cretaceous	opposite	ovate to oblong	no or only at the basal portion	no	petiolate	no	compact female cones with 2 or 3 chlamydosperms	-
	*Gurvanella* [13,64]	Early Cretaceous	opposite	linear	no	no	sessile	no	reduced female cones with bracts having furcate venation and enclosing 2 or 3 chlamydosperms	yes, >3 mm
	*Ephedra* [62,63]	modern	opposite	linear	no	no	sessile	no	reduced compact cones with only the uppermost whorl fertile	yes, 0.5–5 mm
Group IV: female spikes; chlamydosperms verticillate	*Protognetum* [52]	Mid-Jurassic	opposite	linear	no	no	sessile	no	loose spikes with paired leaf-like bracts subtending verticillate chlamydosperms	yes, ca. 0.2 mm long
	*Gnetum* [1,55]	modern	opposite	broad	yes	no	petiolate	no	loose spikes with fused bracts forming annular cupules subtending verticillately arranged chlamydosperms	yes, ca. 0.5 mm
Group V: compact cones; chlamydosperms usually winged	*Welwitschia* [1,65]	modern	opposite	strap-shaped	no	no	sessile	no	compact female cones with modified bracts subtending bilateral chlamydosperms	yes, 4–5 mm

We use “-” to indicate that there are no data/observations of the character.

## Data Availability

All data and materials used in this study are included in this paper.

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
