# Peer review of "A New Gnetalean Macrofossil from the Mid-Jurassic Daohugou Formation"

_plants, 2023, doi:10.3390/plants12091749_

Round 1
Reviewer 1 Report
The paper reports the discovery of an interesting putative gnetalean macrofossil reproductive shoot named Daohugoucladus chinensis gen. et sp. nov. The fossil superficially resembles the current Podocarpus, but characteristics of the female cone showed relationship with other known gnetalean macrofossils. This is a well-written manuscript that provides a thorough analysis of the data. I have carefully reviewed the manuscript and find that it represents a well-executed study that makes an important contribution to the field.
The writing is clear and concise, with a well-structured and informative abstract. Based on my evaluation, I recommend that the manuscript be accepted for publication.
I only have a few minor questions:
Figure 5 and Figure 6, are mp (in figure 5) and mt (in figure 6) the same structure? Why use different abbreviations?
Figure 7 is very interesting. It includes many fossil and living gnetophytes groups and shows evolutionary relationship of their female cones. However, i think that necessary references of each megasporophylls should be cited in the figure legend.
Author Response
- The paper reports the discovery of an interesting putative gnetalean macrofossil reproductive shoot named Daohugoucladus chinensis gen. et sp. nov. The fossil superficially resembles the current Podocarpus, but characteristics of the female cone showed relationship with other known gnetalean macrofossils. This is a well-written manuscript that provides a thorough analysis of the data. I have carefully reviewed the manuscript and find that it represents a well-executed study that makes an important contribution to the field. The writing is clear and concise, with a well-structured and informative abstract. Based on my evaluation, I recommend that the manuscript be accepted for publication.
RESPONSE. Thanks.
- Figure 5 and Figure 6, are mp (in figure 5) and mt (in figure 6) the same structure? Why use different abbreviations?
RESPONSE. We changed “mt” into “mp” for the abbreviation of the micropylar tube projection (page 10, line 227). Thanks for the suggestion.
- Figure 7 is very interesting. It includes many fossil and living gnetophytes groups and shows evolutionary relationship of their female cones. However, i think that necessary references of each megasporophylls should be cited in the figure legend.
RESPONSE. OK, we added references in the figure legends of Figure 7 (pages 15-16, lines 297-307). Thanks for the suggestion.
Reviewer 2 Report
This is a lovely fossil and certainly worthy of study, but I am not convinced it is gnetalean, and think it more likely podocarpalean. I recommend revision, including more detailed comparison with podocarps.
My main misgiving is that I am not convinced that this fossil has been correctly interpreted as verticillate. My view of the structure is that the leaves are on bracteate short shoots. This and the leaf structure makes affinities with Podocarpus much more likely. The structure of the seeds does not really overcome this affinity either, and I could see the outer coat as a shrivelled epimatium.
Minor points
The epithet “chinensis” should be reconsidered. In Latin this would be better “sinensis” (l.21, 86).
l.40, 66 “articulately stated” is awkward> why not just “elaborated”
l.86 add” which is in the Callovian age of the Jurassic”
l.185 spelling should be furrows
l.214 spelling should be lengthy
l.221 delete “This section may be divided by subheadings. It should provide a concise and precise description of the experimental results, their interpretation, as well as the experimental conclusions that can be drawn.”, but do add the subheadings
Author Response
- This is a lovely fossil and certainly worthy of study, but I am not convinced it is gnetalean, and think it more likely podocarpalean. I recommend revision, including more detailed comparison with podocarps.
My main misgiving is that I am not convinced that this fossil has been correctly interpreted as verticillate. My view of the structure is that the leaves are on bracteate short shoots. This and the leaf structure makes affinities with Podocarpus much more likely. The structure of the seeds does not really overcome this affinity either, and I could see the outer coat as a shrivelled epimatium.
RESPONSE. We thought this suggestion carefully and compared our new genus with gnetophytes and Podocarpaceae in a number of characters: 1) leaves are normally opposite in gnetophytes but alternate in Podocarpaceae, our fossil possesses opposite leaves; 2) ovulate organs are orthotropous in gnetophytes, but mostly anatropous in Podocarpaceae; 3) ovules possess a micropylar tube in gnetophytes, but no micropylar tube in Podocarpaceae; 4) previous studies have suggested that the leaf morphology and venation are diverse in fossil gnetophytes (Liu et al. 2013; Yang et al. 2015), we think that the leaf morphology cannot give systematic signals; 5) finally, it is well-known that reproductive characters are important in inferring systematic relationships of plants, our fossil resembles the fossil Siphonospermum simplex from the Early Cretaceous in the pedicled and orthotropous chlamydosperms. We thus consider that this new genus is gnetalean but not Podocarpalean. Thanks a lot for the useful discussion.
Liu, H.M.; Ferguson, D.K.; Li, C.S.; Wang, Y.F. New plants of Gnetales from Early Cretaceous of China, and its bearing on the early evolution of Ephedraceae and Welwitschiaceae in Gnetales. Chin. Sci. Bull. 2013, 58, 200−209.
Yang, Y.; Lin, L.B.; Ferguson, D.K. Parallel evolution of leaf morphology in gnetophytes. Org. Divers. Evol. 2015, 15, 651−662.
- The epithet “chinensis” should be reconsidered. In Latin this would be better “sinensis” (l.21, 86).
RESPONSE. OK, we replaced “chinensis” with “sinensis”. Thanks for the suggestion.
- 40, 66 “articulately stated” is awkward> why not just “elaborated”
RESPONSE. Ok, we replaced “articulately’ with “critically evaluated’. Thanks.
- 86 add” which is in the Callovian age of the Jurassic”
RESPONSE. OK, we added “Callovian”. Thanks.
- 185 spelling should be furrows
RESPONSE. OK, corrected. Thanks.
- 214 spelling should be lengthy
RESPONSE. OK, removed. Thanks.
- 221 delete “This section may be divided by subheadings. It should provide a concise and precise description of the experimental results, their interpretation, as well as the experimental conclusions that can be drawn.”, but do add the subheadings
RESPONSE. We deleted the paragraph (page 10, lines 233-235). Thanks.
Round 2
Reviewer 2 Report
This revision does clarify better the reasons why this is not podocarpalean, but I still find the assumption of verticillate leaf insertion unconvincing. Some of the leaves appear to be on a lateral short shoot.
Author Response
Reviewer#2: This revision does clarify better the reasons why this is not podocarpalean, but I still find the assumption of verticillate leaf insertion unconvincing. Some of the leaves appear to be on a lateral short shoot.
REPLY. We re-examined the type specimen about the possibility of verticillate leaves as mentioned by the Reviewer, and found that the leaves are opposite but not verticillate. The verticillate impression was probably caused by the three broken leaves massed nearby the node. There is no lateral short shoot, but appears to have an axillary pedicle (the dense fine striations associated with the chlamydosperm at the upper right) and three associated broken leaves, one leaf is probably from the curvature of the right leaf of the upper node, and two leaves are associated with the two prominent leaf scars of the node. The phyllotaxy is obvious when seeing the distal nodes. Thanks for this suggestion.

In addition, we replaced “pedunculate” with “pedicled”.